# Wildlife Photos on Social Media: A Quantitative Content Analysis of Conservation Organisations’ Instagram Images

**DOI:** 10.3390/ani12141787

**Published:** 2022-07-12

**Authors:** Meghan N. Shaw, William T. Borrie, Emily M. McLeod, Kelly K. Miller

**Affiliations:** 1Centre for Integrative Ecology, School of Life and Environmental Sciences, Deakin University, Burwood, Melbourne 3125, Australia; b.borrie@deakin.edu.au (W.T.B.); kelly.miller@deakin.edu.au (K.K.M.); 2Department of Wildlife Conservation and Science, Zoos Victoria, Parkville 3052, Australia; emcleod@zoo.org.au

**Keywords:** instagram, conservation, images, conservation organisations, social media, conservation messaging, wildlife

## Abstract

**Simple Summary:**

Although images are more effective than words at communicating important conservation ideas, different aspects of these images have been demonstrated to have positive and negative effects on viewers’ views towards wildlife and towards the organisation that posted the image. The most prevalent and engaging characteristics of wildlife photographs posted to Instagram in 2020 and 2021 were assessed using a quantitative content analysis, with Australian organisations as a case study. The findings show that conservation organisations can confidently share and post photographs that promote positive attitudes towards wildlife and the conservation organisation, and that Instagram posts can feature and promote a wide range of currently underrepresented species.

**Abstract:**

Wildlife populations are vanishing at alarmingly high rates. This issue is being addressed by organisations around the world and when utilizing social media sites like Instagram, images are potentially more powerful than words at conveying crucial conservation messages and garnering public support. However, different elements of these images have been shown to potentially have either positive or negative effects on viewers’ attitudes and behaviours towards wildlife and towards the organisation posting the image. This study used a quantitative content analysis to assess the most common and engaging elements of wildlife images posted to Instagram in 2020 and 2021, using Australian conservation organisations as a case study. A total of 670 wildlife images from the Instagram accounts of 160 conservation organisation Instagram accounts were coded and analysed. Results highlight that the most common image elements used included natural backgrounds, mammals and birds, and no human presence. In addition, it was found that the taxon of the animal featured in a post and the presence of humans did not impact engagement levels. Our findings highlight the potential for Instagram posts to feature and promote a wide range of currently underrepresented species, and for conservation organisations to be able to confidently share and post images that promote positive perceptions of both the animal and the conservation organisation.

## 1. Introduction

Globally, wildlife populations are facing high rates of extinction [1]. More than 1 million species are predicted to become extinct in the next few decades, which in turn could lead to catastrophic changes to the environment and human health and well-being [2]. Numerous conservation organisations around the world are tackling this problem, including not-for-profit organisations, charities, social clubs, zoos, wildlife shelters, rescue organisations, educational institutions, research hubs and private businesses, amongst others [3]. Such organisations can contribute towards conservation efforts through on-ground efforts, such as habitat management, veterinary treatment and wildlife rescue, through to research, education and social outreach [4]. As many of these organisations rely on funding through public donations, volunteers, and government grants, they require strong public support in order to succeed in their conservation efforts [5]. In addition, many aim to change community behaviours to achieve conservation success [6]. Thus, such organisations’ social media profiles need to be carefully managed to ensure that not only are they sending clear and positive messages about themselves, but also about the wildlife that they wish to conserve. 

One way in which many organisations are now promoting themselves, engaging the public in their work and advocating for conservation is by maintaining a social media presence. The use of social media platforms as a means to share information and communicate with others has grown rapidly in recent years [7], with an estimated 44.8 billion users worldwide [8]. Hence, social media is a useful tool for conservation organisations to use to target wide audiences and to engage with their supporters. 

Instagram, a social media platform where users can share and comment on images with short captions, has seen a rapid increase in the number of users and uploads since it was released in 2010, and it is now the most popular photo capturing and sharing app in the world [9]. As a result, many conservation organisations have created accounts on the site as a way to connect with this growing user and supporter base and to share their conservation agendas. 

There is a growing body of research looking at the effectiveness of social media as a way to engage audiences and influence pro-conservation behaviours. Previous research has focused on conservation messaging, including which narratives will foster an organisation’s social license and support [10,11,12]. More recent research has applied these approaches to social media sites such as Facebook and Twitter; however, these have almost solely focused on the text [13,14,15]. For example, analyses of twitter posts have found that threatened species are less likely to be mentioned and discussed than non-threatened species [14], and that the negativity of the discourse behind a species is highly impacted by how it is portrayed in relation to humans [16]. However, there is still much to be learnt about the role of images, which are arguably more powerful at conveying key conservation messages. 

Images have been found to target a deeper level of cognitive processing than text, as their interpretation relies on emotion, memory, and familiarity with the subject matter [17,18]. In particular, the emotions triggered when viewing images can lead to respondents paying close attention to them and remembering their content [17]. However, their interpretation can vary across different audiences, as there is no precise syntax to send a direct and targeted message [19,20]. 

Previous research on wildlife imagery has discovered that elements of an image can impact how viewers perceive the featured animals and the organisations that post the image, and also have an impact on subsequent wildlife-related behavioural intentions [21,22]. 

One example of this is the growing demand for illegal exotic pets, which has been attributed to a rise in social media images featuring such animals as pets. This trend has been identified as contributing to a decline in wildlife biodiversity globally and is a primary threat to many species [23,24,25]. 

Seeing pictures of humans and animals in close proximity may also promote the assumption that wild animals are safe to approach in the wild, which can be dangerous for both animals and humans [26]. Additionally, approaching wildlife can negatively impact the welfare of individual animals [27,28]. Images of humans and animals together have also been shown to impact respondents’ perceptions of the featured animal, and the conservation organisations featured in such images. Subjects who saw photographs of a chimpanzee and a human in close proximity were more likely to think that the animal would make a good pet, according to Ross, Vreeman and Lonsdorf [29]. Similar patterns were found when subjects viewed images of chimpanzees in human settings such as office buildings or wearing human clothing [29].

In addition, Van der Meer, Botman and Eckhardt [22] found that images of humans petting wild cats increased the likelihood that viewers would report intentions to participate in wildlife tourism activities, and that a viewer thought the big cat was not dangerous. However, respondents had more concerns around the featured animal’s welfare when the animal was shown with a keeper or with a member of the public in a zoo setting compared to wildlife tourism and wild settings, highlighting that such images may also impact how organisations that host wild animals may be viewed. Furthermore, Shaw et al. [21] studied the effects of human proximity to animals in wildlife images across a range of taxa, finding that the closer a human is to an animal in an image, the more likely a general audience is to believe that the featured animal would make a good pet. Similarly, close human-animal proximity led to the perception that the animal is not displaying natural behaviours, suggesting that an organisation’s reputation may be affected by the images it posts [21]. 

Research suggests that the species featured within an image impacts how respondents react to the image. Higher levels of engagement and empathy tend to be shown towards species demonstrating some similarity to humans [30]. As such, images of mammals tend to be the most engaging [31,32], due to their forward facing eyes, care for their young, and other human-like characteristics. Viewers have also been shown to prefer colourful and large-bodied animals [33], and as a result, images of birds tend to be more engaging than those of reptiles, fish, and invertebrates. Moreover, viewers tend to prefer younger animals due to a preference for ‘baby schema’, which can elicit strong feelings of empathy and protectiveness [34,35]. 

The background of an image may also promote feelings of connection with an image. For example, naturalistic backgrounds in wildlife images lead to less desire for wild animals as pets than humanistic backgrounds [29,36]. In addition, naturalistic settings and enclosures at zoos have been theorised to increase positive attitudes and emotions towards the species they house, as well as greater value for their survival in the wild [37,38,39,40]. Coloured photographs of wildlife have been shown to increase conservation donations compared to black and white images, highlighting that colour may also have a role in engagement with wildlife photographs [41].

The nature of the image itself has also been shown to have an impact on how viewers respond to it, for example, Oisinski et al. [42] highlighted that cartoon representations of wildlife have the potential to positively impact attitudes towards the featured species and increase pro-conservation behavioural intentions. 

This research on the role of photographs as a conservation messaging tool is relatively new, and as such, there is much still to discover, including the impact of elements such as the number of animals in the frame and the visual qualities of a human that may be in these images. Importantly, although previous research is key in informing appropriate images to aid in conservation promotion, we were unable to locate any research comprehensively examining what kinds of images conservation organisations are currently using, and their success in engaging audiences. 

Australia provides a useful case study to investigate which kinds of images are being shared and their impacts. With some of the highest extinction rates globally, Australian conservation organisations are under pressure to understand how their communications, including their social media posts, can engage the public with conservation objectives [1]. Australia is home to a large array of unique and endangered species, and their conservation relies on the public’s support of the conservation organisations that endeavour to protect them [1]. In addition, 99% of Australians use social media in some form [43], and thus provide a potentially large audience for conservation imagery. 

This study aims to investigate the most common visual elements (e.g., background type, featured species, human presence) in conservation organisation Instagram posts of Australian wildlife. In addition, we investigate which visual elements are linked to higher levels of social media engagement. 

## 2. Materials and Methods

### 2.1. Image Selection

To assess the most common and engaging elements of wildlife images posted by Australian conservation organisations, we assessed a pool of 670 images posted to Instagram in 2020 and 2021. We first define four key terms based on previous literature: (1) Image elements, (2) Wildlife images, (3) Conservation organisations, and (4) Engagement. 

Image elements were defined as the visual characteristics of an image, such as any human, animal or object featured, the background style, and the colour palette, similar to Spooner and Stride [44]. Wildlife images were defined as any image that featured a non-domesticated animal species [45]. Conservation organisations were defined as any group that held a primary goal of promoting conservation and the sustained existence of biodiversity, whether through education, research, rescue, advocacy, or other means [46]. Engagement was defined as the number of likes an image received, divided by the number of followers that the account posting the image had on the day of data collection.

To select the images used for coding, we undertook the following process: 

Firstly, the organisations were selected. We generated a list of all current conservation organisations in Australia by using the search terms ‘conservation’, ‘wildlife’, ‘environment*’, ‘ecolog*’, ‘animal*’, followed by ‘organisation*’ or ‘group*’ and ‘Australia’ on Google, and searched the first 50 pages. Any group that fit the above definition of a conservation organisation was added to a list. This was then supplemented to ensure that organisations missed by this search method, but that were still relevant to our definition, were included. This added registered ZAA accredited zoos in Australia, local councils, national parks, CMAs (Catchment Management Authority), University research groups, wildlife tourism businesses, animal shelters, rescue groups and museums. Once this list had been compiled, the accounts for each were searched for on Instagram. For organisations where it was unclear whether they were based in Australia, their website or ABN [Australian Business Number] was searched to determine their address. 

This list was then further refined upon examination of the Instagram accounts of each organisation. Any account that had not posted in 2021 was removed from the list, followed by removing any organisation that had not posted any images of Australian animals in their last 100 posts. The final number of organisations on this list was 424. 

The list was then sorted into alphabetical order, and each organisation was designated a number that correlated to their order in this list. To ensure a diversity of organisations, a random number generator then generated 160 numbers between 1 and 424, and the organisations that correlated to these numbers were selected as the Instagram accounts from which study images were drawn.

Visiting each of these 160 accounts, the number of Instagram posts featuring images of Australian wildlife between 1 January 2020 and 1 August 2021 were counted, as this covered the timing of the Australian bushfires, a period of ‘normal’ life, and some posts generated during COVID-19 lockdowns in Australia. Posts where multiple images had been included in a slideshow style were not included as it was not clear as to which image viewers may be reacting to. The number of relevant images for each account was then input into a random number generator, and five numbers were drawn. Each number correlated to a relevant image, with 1 being the most recently posted. The images that correlated to each of these generated numbers were then screenshot, and the number of followers of that account was recorded. Selecting multiple images from each account allowed for the sample images to be more representative of the types of images each account posted. This approach produced an image pool of 670 images. This sample size was calculated to provide adequate power with a minimum of 80% power per statistical test. Due to a number of users commenting multiple times on one image, and the large volume of comments some images received, we decided not to include the number of comments in our analysis as it could not provide an accurate measure.

### 2.2. Content Analysis

To investigate these images in detail, a content analysis was undertaken, in which images posted by conservation organisations to Instagram would be coded according to which image elements were present in each. 

A codebook was created to analyse each of the images. Initially, only features that had been mentioned in previous research, such as featured species, human presence and background types were included. The codebook was then further developed from discussions with three wildlife photographers and three social media organisers for conservation organisations, around which elements they believed were the most powerful, and how they created and designed social media images and campaigns. Finally, other image elements that were frequently discussed were added and refined by the coders during the coding training process [S1]. It should be noted that ‘species’ was not always recorded as the exact species, but instead the recognisable name the public may attach to the animal shown, as in Woods [47]. We are interested in measuring viewer responses: viewers being unlikely to discern between, for example, the many bird, lizard, snake and kangaroo species and sub-species, we therefore decided to reflect the more general measure in our study. Additionally, there was a concern that octopuses may elicit different engagement compared to other invertebrates, due to the timing of the release of the film ‘My Octopus Teacher’ in September 2020, hence molluscs were coded as a separate taxon to invertebrates.

Engagement consisted of the number of likes an image received up to (and including) the day it was captured by the research team, divided by the number of followers the account had on that same day, which is based on how Instagram calculates engagement [48]. This measure does not include any calculation of the number of comments received, nor the number of times a story or image was viewed or shared since this information is not publicly available.

The image captions were not included in the coding as the study focuses on the visual elements only. See Figure 1 for an example of the data collected. 

Coding was undertaken by the first author and four additional coders. 

The data were then checked for missing values, with any missing values for each variable subsequently excluded from the analysis. Inter-coder reliability, or the similarity between the five coders (measured as Cohen’s Kappa) was calculated on all images before the final coder resolved any coding disagreements. Cohen’s Kappa was calculated using KappaGUI, an R-shiny app, and yielded a figure of 0.85, regarded as highly reliable [49].

### 2.3. Data Analysis

All data were analysed using linear regressions, analyses of variance, and t-tests in R (R Core Team, Vienna, Austria) and R studio (R Studio Team, Boston, USA). To capture the most frequent category for each categorical variable, frequency tables were calculated. For all continuous variables, including the number of likes, engagement %, number of humans and number of animals, the mean and median values were calculated. Before calculating if there was a significant difference in engagement between categories of each categorical variable, normality was checked using QQ plots. In addition, the homogeneity of variance was calculated using Levene’s test. If the assumptions of normality or homogeneity of variance were violated for a variable (which was the case for background style and animal taxon), Kruskal Wallis tests were used to measure whether there was a significant difference in engagement levels between categories. For continuous variables (number of likes, engagement %, number of humans and number of animals), a linear model was run with the continuous variable as the predictor and engagement as the response variable. For binary variables with normal distributions (whether the image was graphic, and whether a human was touching or holding an animal), we used a Welch two sample t-test. For all other variables (categorical variables with normal distribution and homogeneity of variance), a one-way analysis of variance (ANOVA) test was used. We then used Tukey’s Honestly Significant Different test with Bonferroni corrections to assess the significant differences between categories for each categorical variable.

We recognised that in some instances there were multiple animals or humans within an image, so we included ‘number of animals’ and ‘number of humans’ as interaction terms within all models. However, none of these were significant, and so they were subsequently removed. To further ensure that there was no impact of multiple animals or humans in one image on the analysis, we also ran the same tests with only the images that featured one animal and/or one human. These results were consistent with the original analysis. 

Additionally, we used boxplots to identify significant outliers. We ran the same analyses once more with the outliers removed to assess whether they had an impact on the significance of each test: it was found that they did not. However, they were retained for reporting of the results as they are all valid data points. 

## 3. Results

One of the aims of this study was to describe the common elements used in the Instagram images posted by conservation organisations in Australia. In total, 670 images were included in the final analyses. The majority of images had no text (83%), were photographs (99%) and were of high image quality (52%). Images were also most likely to be an Animal Portrait (93%), feature a naturalistic background (68%), and not be graphic in nature (98%) (Table 1 and Appendix A).

A total of 1120 animals were recorded within the image database. The median number of animals within an image was 1. The majority of animals shown within these images were mammals, shown in a full-body shot, and facing side on to the camera. A large majority were alive, adults and shown at a medium distance from the camera. Most were not brightly coloured (S1). 

147 different species of animals were recorded in the image database. The most common species were the koala, followed by the kangaroo, the lizard, the wallaby, and the Australian King parrot. 

Humans were present in 116 images, however, the overall median count of humans per photo was 0, highlighting that most images did not feature humans. In total, 785 humans were recorded across these 116 images. When humans were present, they were equally likely to be male or female. They were also more likely to be an adult, considered a member of the general public, and the shot was more likely to show only their hands or arms, instead of their face. The majority of featured humans were touching an animal; however, most were not feeding the animal. Roughly half of the humans were holding the animal. When a human was shown in an image, they were also likely to be shown close to an animal. 

Most of the variables did not have a significant relationship to engagement, including the time period the image was posted in (F(2, 667) = 1.92, *p* = 0.15). For a full list, see Appendix A. The variables that did significantly impact on image engagement were whether there was text on the image, the image quality, and the taxon of the animal in the image. We provide partial eta squared and Cohen’s *d* to indicate effect sizes. 

### 3.1. Text on the Image

A significant relationship was found between the presence of text on an image, and the level of engagement it received (t = −3.012, df = 169.95, *p*< 0.001, d =0.30). Images with no text received higher engagement (5.3% compared to 3.8%). Although this change appears small, a very high engagement rate is considered to be anything above 6%, thus even a small percentage change has a measurable impact [50] (Figure 2).

### 3.2. Image Quality

Homogeneity of variance was assured by a Levene’s test (F = 0.25, df = 3.0, *p* = 0.8), and an Analysis of Variance (ANOVA) revealed a significant relationship between image quality and engagement, with poor quality images receiving higher engagement than high quality images (High = 4.5%, Medium = 5.2%, Low = 6.0%, Poor = 6.7%) (F(3, 660) = 3.392, *p* = 0.02,ηp^2^ = 0.02) (Figure 3).

### 3.3. Taxon

The differences between taxon categories did not display homogeneity of variance (F = 2.92, df = 6, *p* = 0.008). However, the Kruskal Wallis test highlighted that the taxon shown in an image still had a significant impact on the engagement an image received, though the effect size was very small (H= 32.94, df = 6, *p* < 0.001, ηp^2^ (H) = 0.003). In particular, the engagement level for mammals was significantly higher than that for invertebrates, birds, or reptiles (Invertebrate = 3.2%, Mollusc = 5.0%, Fish = 4.2%, Bird = 4.4%, Mammal = 6.1%, Amphibian = 3.8%, Reptile = 4.1%) (Figure 4).

### 3.4. Touching and Holding

Higher engagement was also recorded when a human was touching (t = 2.72, df = 148.71, *p* = 0.007, d = 0.45) or holding (t = 2.91, df = 101.84, *p* = 0.004, d = 0.58) an animal, however when outliers outside of the interquartile range were removed, these statistical results were no longer significant.

## 4. Discussion

Our analysis of Instagram images posted by conservation organisations demonstrates a great variety of image styles, of elements in those images, and the presence or absence of humans (Table 1). The results also suggest that most image elements when analysed separately do not significantly impact viewer engagement. This finding highlights that conservation organisations can design and construct their images in a way that promotes conservation and animal welfare, and sends appropriate messages about the animal and the organisation, without compromising viewer engagement. Indeed, contrary to our hypothesis, the main variables that significantly impacted the engagement with an image were related to image characteristics instead of to the features of the animal or human within. This may suggest that the way the image is framed, edited and posted may have just as much of a role in engagement as the content within. 

### 4.1. Image Characteristics 

For example, images that contained text were found to be less engaging. The human brain can process an image and begin to understand meaning within 13 milliseconds, which is 60,000 times faster than text [51]. With the average user only spending 1–2 s on each post [52], text can often be overwhelming and these posts can be skipped due to the mental work they require. Instead, images that focus on the animal instead of containing text may encourage more engagement. As has been demonstrated in other fields [53,54], research should expand upon the role of text on, and as captions to, images posted on Instagram, to uncover how these may work in tandem to elicit engagement and impact perceptions of conservation-related issues. 

In addition, the quality of the image significantly impacted engagement levels, with lower quality images receiving more engagement. We suggest that this could be due to the authenticity that lower-quality images may convey. Perhaps, as the average person is now more likely to own a phone camera than a high-quality DSLR, lower-quality photographs may highlight the ‘unexpected’ and chance encounters that viewers can have with wildlife, as compared to scenarios with professional cameras that may appear more engineered. Viewers are more likely to engage with images in which they can ‘transport’ themselves, or see themselves within [55]. In this case, lower-quality images may be more relatable, and demonstrate an experience the viewer could recreate. This finding was not anticipated, and we recommend that future research should investigate the mechanisms behind why this result occurred.

Although they did not impact engagement, it is interesting to note that the majority of images were animal portraits, defined by our research team as images that featured an animal as the focal point (e.g., no humans in the frame, and the animal was distinct from the background). This highlights that when conservation organisations are posting images of wildlife, they are often the key subject of these images. A focus on the animal can promote feelings of kinship and connection with the animal, which in turn can elicit support for its conservation [56]. 

### 4.2. Animal Elements

The only Animal Element to have a statistically significant relationship with engagement was the taxon. This may also be the case for species, but could not be measured adequately due to lower counts for most species. In agreement with our hypothesis, mammals were found to be significantly more engaging than birds, reptiles or invertebrates. This reflects the literature on species preference, which has shown that large-bodied, fluffy animals with forward-facing eyes are more likely to be preferred by members of the public [57,58]. In fact, a study undertaken on Australians’ favourite species found that 8 out of 10 of the top species were mammals [47]. We must also recognise that Australia has a wide range of unique mammals that cannot be found elsewhere, particularly marsupials and monotremes. The uniqueness of a species can also be a key factor in species preferences, with more unique species often preferred [47,58,59]. Additionally, many Australian mammals are well recognised by the public and used as flagships to promote both conservation and Australian identity [10,60]. Emotional attachments to both the species and the cause it represents may also explain why mammals were found to be more engaging than other taxa. However, separating molluscs from invertebrates in taxon analyses showed images of invertebrates had differences in engagement levels compared to mammals, but molluscs did not. This may be a function of the film ‘My Octopus Teacher’, which was highly successful around the time of the study, whilst eliciting empathy for and connection to molluscs [61].

Nevertheless, mammals were more likely to be posted on Instagram than other taxa, perhaps due to an underlying perception that they are more engaging. As a consequence, other taxa such as invertebrates, amphibians and molluscs are generally underrepresented in posts. Such underrepresented species often find themselves at a higher risk of extinction due to a lack of conservation support over their better-known counterparts [60].

However, our findings also highlight that apart from mammals, there are no significant differences in engagement levels by which taxon is featured in an image. The very small effect size of this significant difference should also be considered. We argue that this provides evidence and encouragement for conservation organisations to represent and promote a wider range of taxa and species in their social media posts. Conservation marketing work by Curtin and Papworth [62] and Veríssimo et al. [63] has highlighted that increased representation and marketing efforts for underrepresented, and even unappealing, species can shift conservation support towards these species, and elicit greater donations for them. In addition, the theory of repeated messaging [64] suggests that the more a person is exposed to a message, or in this case, species, the more likely they are to recognise it and support it. Consequently, the lack of major engagement differences between different taxa provides the opportunity for conservation organisations to expose viewers to a wider diversity of species and in turn, to encourage support of biodiversity. 

Furthermore, most animals were not shown looking at or even facing the camera. Although this can assist in increasing feelings of kinship and connection with the animal, we recognise that many conservation organisations post imagery of wildlife where the animal’s position is not controlled by the photographer and that this models appropriate and responsible behaviours for wildlife photography [65]. In addition, we recognise that many photographers will use the zoom function to make the animal appear closer to the camera than it actually is. However, we also recognise the impact these images may have on viewers’ perceptions of what is an appropriate distance to keep from wildlife. For example, previous studies have found that when certain behaviours are shown in social media posts, they are more likely to grow in popularity [66]. In addition, when viewing wildlife, people prefer to be as close as possible to the animal [67]. Hence, when shown these behaviours on social media, the public may see their preference as acceptable. Conservation organisations can use the findings from this study to trust that they can post images of animals at safer distances without being concerned that their posts will not engage their audience. In addition, posts that do feature close-ups of animals may wish to include information in the caption about the type of zoom or lens used. Future research could investigate the role of an animal’s proximity to the camera in photographs on the human—wildlife proximity behaviours of viewers to determine how strong this influence is.

### 4.3. Human Elements

Humans were only shown in ~17% of images, highlighting that there is a low risk of the negative effects of viewing humans in wildlife imagery impacting a large proportion of the population. Moreover, the only human-related variables that had a significant relationship with engagement were when a human was shown interacting with an animal, either by touching or holding them, although this was impacted by outliers. Building on the theory of ‘transportation’, an explanation for why images of humans touching and holding animals can be more engaging may be that viewers can imagine themselves as the human model, having that experience for themselves [68]. This is supported by van de Meer et al. [22], who found that when humans were shown with big cats in photographs, viewers were more likely to want to participate in a wildlife tourism experience with a big cat. In addition, viewers are often drawn to human faces in images, hence images featuring humans are thought to be more engaging [69]. Many of the human—wildlife images in our sample did not show the human’s face, which may also assist in feelings of ‘transportation’, as the viewer cannot see as many physical differences between themselves and the featured human. 

However, human presence had little or no impact on the engagement an image received, highlighting that it is likely to be the interaction the human had with the animal that is drawing attention rather than the human themselves. Previous research has shown that a human’s presence, proximity to, and interaction with an animal in a photograph can increase a viewer’s perception that the animal is not endangered, would make a good pet, and is not able to display natural behaviours; all these are detrimental messages for conservation organisations to send about their organisation and the animal they’re featuring in the image [21,29]. Accordingly, conservation organisations can feel confident that they do not need to feature humans with wildlife in their posts to elicit engagement, so do not need to risk the unintended messages that human—wildlife photographs can send. 

We note that for both the touch- and hold-interactions, results were skewed by extreme outliers; some images of human—wildlife interactions received extraordinarily high levels of engagement in the sample. When these were removed for exploratory analysis, neither variable had a significant relationship with engagement. Thus, such interactions may not produce reliable levels of engagement and may not be necessary to sustain an Instagram audience. 

Overall, this study is an initial exploration of the roles of photographic elements of wildlife photographs in isolation from each other. However, we suggest that it is unlikely that engagement is impacted by these elements individually, and instead that it is likely to be a combination of these elements that has a stronger impact on how much engagement an image elicits. It is also unlikely that the perceptions viewers draw from these images are impacted by just one element in isolation, although this is often how such studies are undertaken. We recommend future studies research the impact of a variety of image elements on viewers taken together, to begin to tease apart the underlying relationships between various image elements and their impact on viewers. 

We recognise that likes alone are not a fully comprehensive measure of engagement, as viewers may spend time viewing and considering images without liking them. Unfortunately, we did not have access to each account’s metadata and metrics, which includes information on how long an audience looks at an image, how many people shared the image, and other such details which may provide a more accurate measure of the actual engagement with an image. Future studies may wish to source these from account owners for a clearer understanding of engagement. Future studies may also wish to include comments in their engagement calculations. We were unable to use comments in our calculations in light of certain followers commenting numerous times on the same image. However, our measure for engagement is certainly effective not only in highlighting engagement, but also the conscious choice made by account followers to like a post, thus engaging a deeper level of cognitive processing such that viewing the image affects an actual behaviour. Importantly, certain conservation organisations with their own Instagram accounts may have different results from our findings and may find certain images are more engaging than others to their specific audiences. Therefore, we highlight that our results are general in nature, and recommend that conservation organisations still make informed choices on the images they post based on their messaging goals and specific circumstances.

Lastly, it is likely that many of the followers of the accounts were already interested in conservation or wildlife, as they had made the active choice to follow conservation organisations on Instagram. Consequently, our findings may or may not relate to a broader audience, and we recommend that further research on image engagement be conducted with a sample of the general public. This is of particular importance for campaigns and marketed posts that are aiming to reach people outside of a conservation organisation’s usual reach. 

## 5. Conclusions

Our study highlights that conservation organisations can post responsible wildlife imagery without having to be concerned about some commonly anticipated responses an image may invoke. Encouragingly, many of the elements of wildlife imagery that may lead to negative implications, such as human presence and gore, are not commonly used. Therefore, the negative effects are likely to not be far-reaching. In addition, although some common image trends such as human—wildlife photographs may have a negative impact on wildlife, our research provides encouragement that such images are not necessary for audience engagement. Our findings highlight the potential for Instagram posts to feature and promote a wide range of currently underrepresented species, and for conservation organisations to be able to confidently share and post images that focus on promoting positive perceptions of both the animal and the conservation organisation. 

## Figures and Tables

**Figure 1 animals-12-01787-f001:**
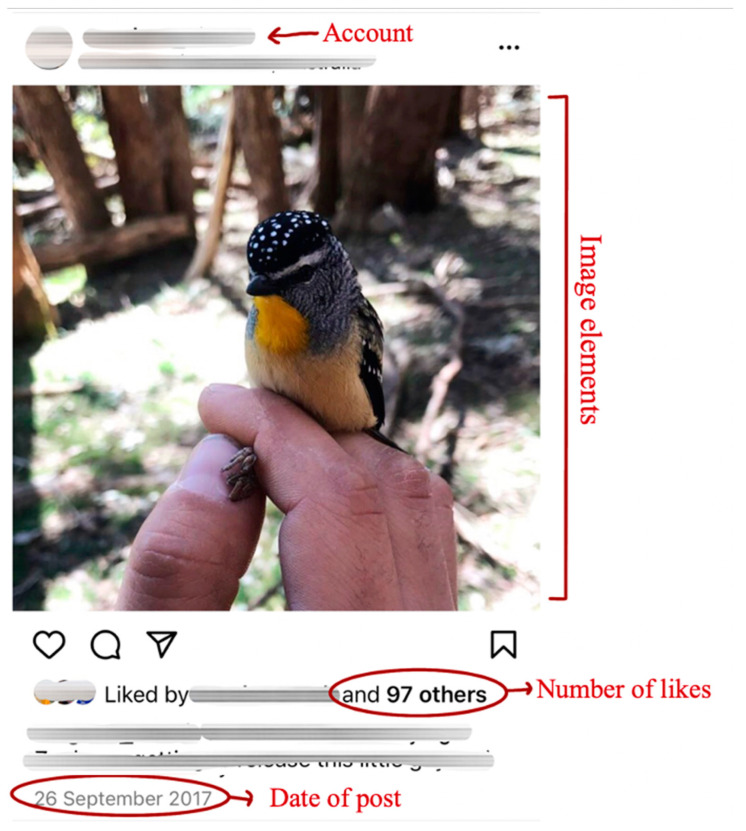
Example Instagram post and the types of data collected (in red) M. Shaw (2015).

**Figure 2 animals-12-01787-f002:**
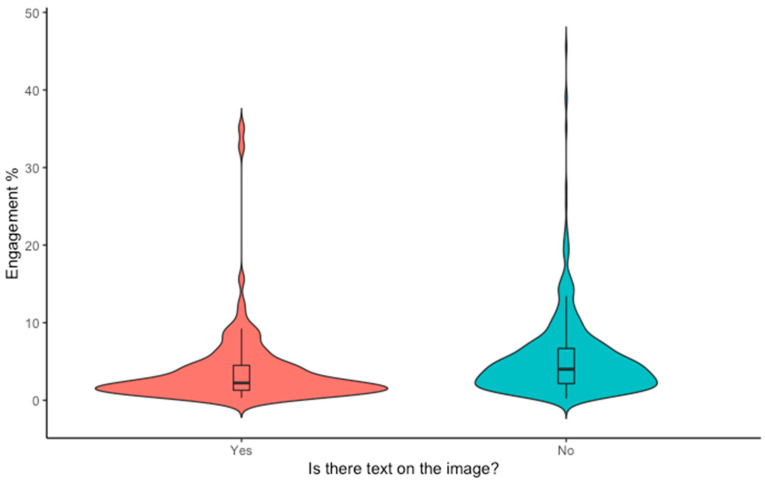
Violin plot with mean and interquartile range (IQR) showing the relationship between text on an image and viewer engagement (%) for Australian wildlife images posted to Instagram by conservation organisations from 1 January 2020–1 August 2021. Engagement is calculated as the number of likes an image received divided by the account follower count.

**Figure 3 animals-12-01787-f003:**
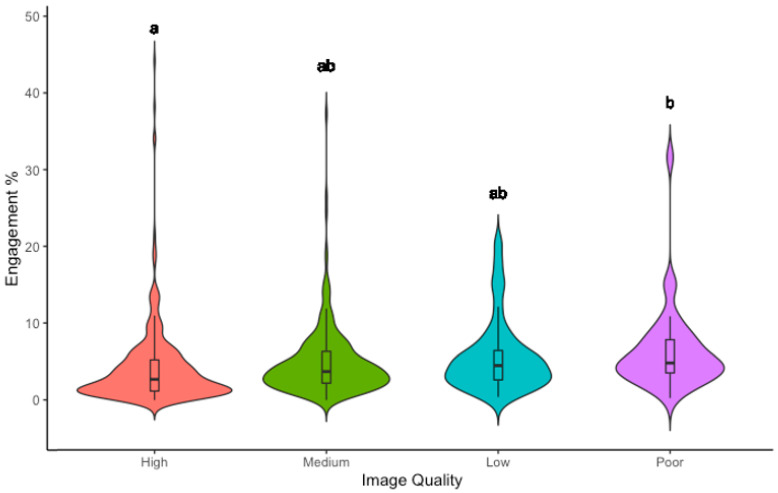
Violin plot with mean and interquartile range (IQR) of the relationship between Image Quality and Engagement (%) for Australian wildlife images posted to Instagram by conservation organisations from 1 January 2020–1 August 2021. Significant differences between categories are denoted by lettering, with the same letter representing no significant difference. Engagement is calculated as the number of likes an image received divided by the account follower count.

**Figure 4 animals-12-01787-f004:**
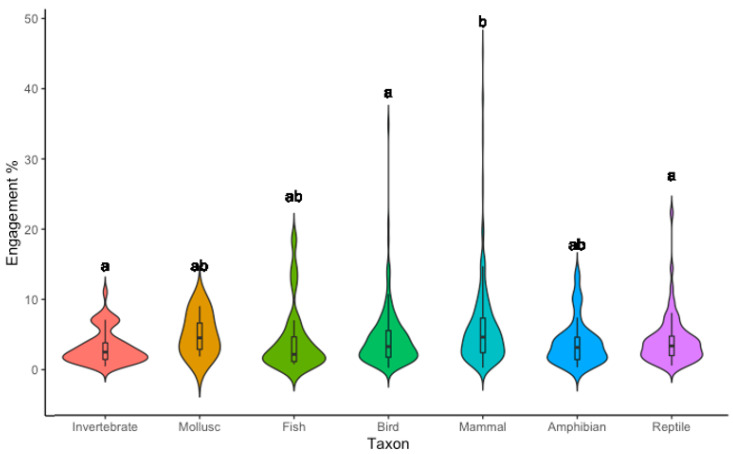
Violin plot, mean and interquartile range (IQR) of engagement levels (%) for each Taxon shown in Australian wildlife images posted to Instagram by conservation organisations from 1 January 2020–1 August 2021. Significant differences between categories are denoted by lettering, with the same letter representing no significant difference. Engagement is calculated as the number of likes an image received divided by the account follower count.

**Table 1 animals-12-01787-t001:** The frequency of elements in a sample of 670 wildlife images posted on Instagram by Australian conservation organisations.

Variable	Category	Definition	Count	%
**When was** **the image** **taken** **?**	Australian Bushfires	Between 1 January 2020 and 22 March 2020	53	0.6
	Australian COVID-19 Lockdowns	Between 23 March 2020 and 1 May 2020, between 8 July 2020 and 27 October 2020 and between 24 April 2021 and 1 August 2021	355	4.1
	Normal life	Any other time outside of these dates	8302	95.3
**Is there text on the image?**	Yes	There is text on the image	116	17.3
	No	There is no text on the image	553	82.7
**What type of image is it?**	Photo	A photograph	663	99.1
	Cartoon	A cartoon [2D image]	1	0.2
	Illustration	A hand-drawn image	4	0.6
	Computer drawing	A 3D image that is not a photograph	1	0.2
**What is the image quality/resolution?**	High	Images thought to be taken with a DSLR style camera	344	51.8
	Medium	Images thought to be taken with a higher quality phone or digital camera	242	36.5
	Low	Images thought to be taken with a lower quality or older style camera	41	6.2
	Poor	Images with very low resolution and/or blurring	37	5.6
**What style is the image?**	Animal Portrait	An image that has an animal as the focal point	622	92.8
	Landscape	An image that has the background as a focal point	20	3.0
	Human Selfie	An image that has a human as the focal point	27	4.0
	Other		1	0.2
**What is the background of the image?**	Naturalistic	A background featuring nature and natural elements	452	67.8
	Human Made	A background featuring human-made environments or elements—such as a house, street, vet clinic etc.	171	25.6
	Blank	A plain background, generally white or black	31	4.6
	Other		13	1.9
**Is there graphic content in the image?**	Yes	The image features gore or dead animals	11	1.6
	No	The image does not feature gore or dead animals	659	98.4
**What is the taxon of the animal?**	Invertebrate		70	7.9
	Mollusc		11	1.2
	Fish		34	3.8
	Bird		261	29.4
	Mammal		394	44.3
	Amphibian		32	3.6
	Reptile		87	9.8
**Where is the animal facing?**	Facing camera	Facing camera but not making eye contact with the camera	279	31.4
	Back to camera	Back of head is facing camera	84	9.4
	Side to camera	Side of head is facing camera	397	44.6
	Face not visible	Face is not in the image—e.g., is cropped out	18	2.0
	Looking at camera	Facing camera and making eye contact with the camera	112	12.6
**Distance of animal from the camera**	Distant	Appears more than 10 m away from lens	13	1.5
	Far	Appears ~5–10 m away from lens	80	9.0
	Medium	Appears ~2–5 m away from lens	299	33.6
	Close	Appears ~1 m away from lens	445	50.0
	Very Close	Appears less than 30 cm away from lens	53	6.0
**Is the human touching an animal?**	Yes		100	66.2
	No		51	33.8
**Is the human holding an animal?**	Yes		75	49.7
	No		76	50.3
**What is the distance of the human from the animal?**	Touching		98	67.6
	Close ~30 cm		24	16.6
	Far ~1 m		15	10.3
	Very Far ~5 m+		8	5.5

## Data Availability

The data presented in this study are openly available in Figshare at https://doi.org/10.6084/m9.figshare.20280246 (accessed on 4 April 2022).

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
