# Peer review of "Wildlife Photos on Social Media: A Quantitative Content Analysis of Conservation Organisations’ Instagram Images"

_animals, 2022, doi:10.3390/ani12141787_

Round 1

Reviewer 1 Report

The use of social networks for wildlife conservation is gaining increasing importance. This work showed that photos uploaded on social networks may improve social attitude towards biodiversity.

I have a few revisions before final acceptance

1.      Authors of reference n 13 are not correct.

2.      References should be placed in “square brackets”.

3.      LINE 37. “Extraordinary” may not be the right term. Please clarify.

4.      Why did you sample for less than 1 year? Please be clear.

5.      Line 236. Are you sure that you have verified all the assumptions to perform these tests?

6.      Line 278. The kangaroo, the lizard and the wallaby are not species, but species groups. Which species of kangaroo, lizard and wallaby?

7.      Line 280. with a median count of 0 humans per image, what do you mean?

Reviewer 2 Report

Shaw et al. have conducted an interesting investigation of social media image content uploaded by conservation organizations to investigate which image elements are most related to engagement by followers. There is certainly value in this approach to help guide conservation organizations in engendering positive engagement and attention for conservation. 

The manuscript is thoroughly written and well-researched. My comments are mostly minor and are mostly associated with justifying sample sizes decisions and clarifying statistical analyses and results. Overall I think this is an interesting and valuable contribution.

L183: Why subset this? I'm sure it was done for logistical reasons but state that. How was 160 decided upon? Power analysis? If not, I suggest doing so to ensure sufficient statistical power.

L190: Any fear of bias associated with these major historical time periods? Was perhaps engagement overall just higher during the pandemic because people had more screentime? I think this should be explored a bit and if qualitative investigations of the data support differences between the three major time periods (pandemic, bush fires, normal life) this could be accounted for as a random effect in analysis with relatively little change analyses.

L196: Same question as above - how were these sample sizes decided upon?

Figure 1: This shows that comments were extracted for analyses, but around line 200 you state that you decided not to use comments as they were unreliable. Adjust figure accordingly

Figure Captions: Suggest expanding all figure captions to provide more information about the scope and timing of your study so that they could be standalone without the manuscript

L230-231: what does this mean?

Section 2.3: all of these analyses seem correctly applied and appropriate but the presentation style makes that difficult to judge. They are all mentioned at once without pairing the particular analyses to the particular response variables. Suggest re-writing this section to better inform the reader which types of analyses were used for which data

Table 1: Formatting is messed up from the "Graphical Content" section onwards. Re-adjust sections

Invertebrate not listed under species categories. Why was mollusc separated from invertebrate?

L278: kangaroo

Section 3.2 and onward: Now sure why "ηp2" was included. I don't think I've seen this reported before.

Round 2

Reviewer 1 Report

Authors have amended the MS following all of my previous suggestions, therefore I suggest the paper to be accepted for publication.